# A DNA-Modified Live Vaccine Prime–Boost Strategy Broadens the T-Cell Response and Enhances the Antibody Response against the Porcine Reproductive and Respiratory Syndrome Virus

**DOI:** 10.3390/v11060551

**Published:** 2019-06-14

**Authors:** Cindy Bernelin-Cottet, Céline Urien, Elisabeth Stubsrud, Virginie Jakob, Edwige Bouguyon, Elise Bordet, Céline Barc, Olivier Boulesteix, Vanessa Contreras, Christophe Barnier-Quer, Nicolas Collin, Ivan Trus, Hans Nauwynck, Nicolas Bertho, Isabelle Schwartz-Cornil

**Affiliations:** 1VIM, INRA, Université Paris-Saclay, Domaine de Vilvert, 78350 Jouy-en-Josas, France; bernelin.cottet.cindy@hotmail.fr (C.B.-C.); celine.urien@inra.fr (C.U.); edwige.bouguyon@inra.fr (E.B.); elise.bordet02@gmail.com (E.B.); nicolas.bertho@inra.fr (N.B.); 2Vaccibody AS, Gaustadalleen 21, 0349 Oslo, Norway; EStubsrud@vaccibody.com; 3Vaccine Formulation Laboratory, University of Lausanne, Chemin des Boveresses 155, 1066 Epalinges, Switzerland; virginie.jakob@unil.ch (V.J.); barnierquer@gmail.com (C.B.-Q.); Nicolas.Collin@unil.ch (N.C.); 4Plate-Forme d’Infectiologie Expérimentale-PFIE-UE1277, INRA, 37380 Nouzilly, France; celine.barc@inra.fr (C.B.); olivier.boulesteix@inra.fr (O.B.); 5Immunology of viral infections and autoimmune diseases, IDMIT Department, IBFJ, INSERM U1184-CEA-Université Paris Sud 11, 92260 Fontenay-Aux-Roses et 94270 Le Kremlin-Bicêtre, France; vanessa.contreras@cea.fr; 6Laboratory of Virology, Faculty of Veterinary Medicine, Ghent University, Salisburylaan 133, B-9820 Merelbeke, Belgium; ivan.trus@gmail.com (I.T.); hans.nauwynck@UGent.be (H.N.)

**Keywords:** PRRSV, DNA vaccine, modified-live vaccine, antigen-presenting cell targeting, pigs

## Abstract

The Porcine Reproductive and Respiratory Syndrome Virus (PRRSV) induces reproductive disorders in sows and respiratory illnesses in growing pigs and is considered as one of the main pathogenic agents responsible for economic losses in the porcine industry worldwide. Modified live PRRSV vaccines (MLVs) are very effective vaccine types against homologous strains but they present only partial protection against heterologous viral variants. With the goal to induce broad and cross-protective immunity, we generated DNA vaccines encoding B and T antigens derived from a European subtype 1 strain that include T-cell epitope sequences known to be conserved across strains. These antigens were expressed either in a native form or in the form of vaccibodies targeted to the endocytic receptor XCR1 and CD11c expressed by different types of antigen-presenting cells (APCs). When delivered in skin with cationic nanoparticles and surface electroporation, multiple DNA vaccinations as a stand-alone regimen induced substantial antibody and T-cell responses, which were not promoted by targeting antigens to APCs. Interestingly, a DNA-MLV prime–boost strategy strongly enhanced the antibody response and broadened the T-cell responses over the one induced by MLV or DNA-only. The anti-nucleoprotein antibody response induced by the DNA-MLV prime–boost was clearly promoted by targeting the antigen to CD11c and XCR1, indicating a benefit of APC-targeting on the B-cell response. In conclusion, a DNA-MLV prime–boost strategy, by enhancing the potency and breadth of MLV vaccines, stands as a promising vaccine strategy to improve the control of PRRSV in infected herds.

## 1. Introduction

The Porcine Reproductive and Respiratory Syndrome Virus (PRRSV), a positive-strand RNA virus of the Arteriviridae family, induces respiratory illness in neonatal and in recently-weaned piglets as well as late abortions/early farrowing in sows and is responsible for major economic losses in the swine industry worldwide [1]. This virus, which primarily infects only certain types of macrophages including pulmonary alveolar macrophages, exists in two distinct species, PRRSV-1 and PRRSV-2, which are the dominating species in Europe and North America, respectively [1,2]. Phylogenetic analyses show that PRRSV-1 includes four different subtypes and ORF2–5, which encode for viral glycoproteins, present around 30% substitutions at the amino-acid level across strains [3]. The PRRSV genome undergoes frequent random mutations and intragenic recombination, generating new variants responsible for pathogenic outbreaks [3]. The most effective commercial vaccines are modified live vaccines (MLVs), attenuated by successive in vitro passages. However, MLVs revealed safety concerns with reversion to pathogenicity and only demonstrated partial protection against heterologous strains. This limited cross-strain protection is considered as responsible for the failure of the current commercial MLV vaccines to control PRRSV infections in endemic zones [4,5,6].

The protective immune effectors against PRRSV involve neutralizing antibodies, which are mostly directed to variable antigenic determinants of outer membrane proteins [7]. However, protective immunity can be obtained in the absence of neutralizing antibodies and T-cell responses have been proposed to be protective effectors [8,9,10,11]. Indeed, IFNγ is effective at inhibiting PRRSV replication [12], and the IFNγ T-cell responses were correlated to viremia and viral clearance [8,9]. Interestingly, both CD4^+^ and CD8^+^ T-cells can exert anti-viral effects on infected alveolar macrophages expressing MHC class I and II molecules loaded with viral peptides. These viral peptides can presumably originate from all viral molecules, including internal structural and non-structural proteins, which are more conserved than the external ones. Furthermore, the T-cell receptor recognition may tolerate viral peptide mutations, provided that the overall electric charge and peptide–MHC complex shape is maintained [13]. Therefore, a so-called “T-cell vaccine” may stand as an interesting control tool against PRRSV. Finally, by using a PRRSV-1 proteome-wide synthetic peptide library, several T-cell peptides were found to present a good degree of conservation across PRRSV-1 strains and pigs infected with distant strains showed a remarkably similar profile of T-cell antigen-reactivity [14,15]. The most frequently recognized and conserved T-cell antigen sequences were found in NSP5, NSP1β, RdRp, M and GP5 [14,15,16].

DNA vaccines are particularly efficient at stimulating T-cell responses [17]. In addition, a huge body of literature indicates, yet mainly in mouse models, that targeting antigens to the most efficient antigen presenting cell (APC) type, i.e., conventional dendritic cells (cDCs), strongly enhances the magnitude and breadth of T-cell responses [18,19,20,21], including in the case of DNA vaccination [22,23,24,25]. With the goal to trigger broad T-cell responses against PRRSV with DNA vaccination, we expressed PRRSV antigens (PRRSV-AGs) containing conserved T-cell epitopes in vaccibody structures (VB), which are dimeric platforms designed to target antigens to cellular receptors [23,24,25], being here pig XCR1 and CD11c. Indeed, XCR1 is selectively expressed by the cDC1 type, which excels in Th1 and CD8^+^ T-cell activation, as found in the mouse, human, monkey, sheep and pig [26,27,28]. CD11c, which is broadly expressed on APCs including cDCs and macrophages in pigs [29], has successfully been used to induce Th1 and CD8^+^ T-cell responses in mouse [30,31]. In human, its intracellular routing to early endosomes was proposed to favor cross-presentation to CD8^+^ T-cells [32] and we showed its potential to promote IFNγ T-cell responses in pigs against the influenza NP protein [33]. We evaluated the immunogenicity of DNA constructs encoding PRRSV-AGs targeted to APCs or in native forms, as stand-alone vaccines or in a prime–boost strategy with an MLV vaccine. The DNA vaccines were combined to cationic nanoparticles (NPs) and delivered in skin with surface electroporation (EP). Indeed, the combination of DNA with NPs and EP has been previously shown to be most efficient at triggering efficient transfection and immunogenicity, possibly by favoring the intracellular routing of the DNA to the nucleus and/or its sensing in the transfected cells [34] (and our unpublished results, see companion manuscript [35]). Surface electroporation in the skin is expected to lead to transfection of a large variety of skin cell types [36,37], and the expression of the vaccine antigen, especially when targeted to APCs, is ideally located to favor the capture by skin APCs, whose density is quite high in this organ. With this strategy, we found that the DNA-MLV prime–boost regimen induced broader T-cell responses and higher antibody responses as compared to DNA alone or MLV alone and that APC-targeting to CD11c and XCR1 only favored the antibody response induced by the DNA-MLV prime–boost strategy.

## 2. Material and Methods

### 2.1. Antibodies

The anti-pig IFNγ mouse P2G10 mAb (capture) and biotinylated anti-pig IFNγP2G11 mAb were from MabTech AB (Nacka Strand, Sweden). The anti-pig IgA mouse mAb K61-1B4 was bought from BIO-RAD Antibodies (Hercules, CA, USA) and the HRP-conjugated rat anti-mouse IgG1 from BD-Biosciences (San Jose, CA, USA). The hybridoma producing the murine anti-pig CD11c monoclonal antibody (mAb) has been previously described [29]. In flow cytometry experiments, the used mAbs are the murine anti-pig MHC class II mAb (clone MSA3, IgG2a, Washington State University Monoclonal Antibody Center (WSU), Pullman, WA, USA), the anti-pig CD172a mAb (clone 74-22-15A, IgGb, WSU, USA), murine isotype control mAbs, the murine anti-CH3 domain of human IgG mAb (clone 57H, IgM, BIO-RAD antibodies) and the biotinylated murine anti-CH3 domain of human IgG mAb (biot-anti-CH3, clone HP6017, IgG2a, ThermoFisher Scientific, Waltham, MA, USA). The conjugated isotype-specific antibodies were bought from ThermoFisher Scientific and are: Alexa 647-conjugated (A647) goat anti-mouse (GAM) IgG2a, APC-cy7-conjugated (APC-cy7)-GAM IgG2b, and A488- GAM IgM.

### 2.2. Modified Live Virus

The PRRSV-1 subtype 1 13V091 strain [38] (FL13) was attenuated by 73 in vitro passages in the MARC-145 cell line, clone F4. The attenuated strain was named 13V091b (designated as FL13b in this paper) and was unable to infect primary alveolar macrophages, even after in vivo passage.

### 2.3. Constructions of Vectors Expressing PRRSV-AGs

pcDNA3.1 vectors encoding four PRRSV-AG sequences from the FL13 isolate, PRRSV-1 subtype 1, were synthetized by GeneArt (InVitrogen, ThermoFisher Scientific,). The PRRSV-AG sequences are: (i) a GP4GP5M chimera that was derived from a previous work on a PRRSV-2 virus [39] and transposed to the FL13 virus (1008 nt); (ii) NSP1β (1176 nt); (iii) RdRp (1962 nt); and (iv) N (405 nt). In the case of NSP1β and RdRp, which are naturally synthetized from viral polyprotein precursors (ORF1 and -2), an exogenous kozak sequence was added. Only GP4GP5M is expected to be secreted as it includes the natural leader peptide of GP4. Each antigen sequence was terminated by a 6× His tag. The plasmids are named pGP4GP5M, pN, pNSP1β, and pRdRp.

The construction of the VB vaccine format has been previously described [40]. The VB format includes a targeting unit, a dimerization unit made of a human Ig hinge and CH3 sequences of human IgG3, and an antigen unit (Figure 1). A VB with porcine XCL1 as the targeting unit has been previously published by our group [29] and this same targeting unit sequence including the natural XCL1 leader sequence was used here. The CD11c targeting unit is a single chain Fragment variable (scFv) derived from the mouse IgG1 hybridoma 3A8 anti-pigCD11c whose sequence was determined by Synogene (San Diego, CA, USA). The sequence was deposited on Bankit (https://www.ncbi.nlm.nih.gov, BankIt2157014 scFv, MK033322, to be released on February 2020). The leader sequence used in the CD11c-targeted VB was derived from a mouse Ig (MNFGLRLIFLVLTLKGVQC). The four PRRSV-AGs used as antigen units (see above) were checked to be free of membrane retention motives using the Annie prediction tool [41]. All sequences were codon-optimized according to the porcine species, synthesized by GenScript (Piscataway, NJ, USA) and cloned in a pUMVC4a vector (Aldevron, Fargo, ND, USA). NSP1β was split into two parts, from AA1-202 (NSP1β1) and AA182-383 (NSP1β2). The generated VB plasmids with successful expression and secretion properties (see Section 3) used in this study are named pXCL1-N, pXCL1-GP4GP5M, pXCL1-NSP1β1, pXCL1-NSP1β2, pscCD11c-N, and pscCD11c-GP4GP5M. Plasmid productions for immunization were prepared using endotoxin-free NucleoBond^®^ EF kit (Nagel Macherey GmbH, Düren, Germany) according to the manufacturer’s instructions and were stored at −20 °C until use.

### 2.4. 293T Transfection

293T cells were cultivated in DMEM with 10% FBS (Eurobio Ingen, Courtaboeuf, France). They were transfected with plasmids encoding the vaccine constructs by use of Lipofectamine 2000 following the supplier recommendations (Invitrogen, ThermoFischer Scientific). A negative control with an irrelevant pcDNA3.1 plasmid was used (Invitrogen, ThermoFischer Scientific). Twenty-four-hours post-transfection, the culture medium was replaced with serum-free medium (Pro293E-CDM, Lonza, Basel, Switzerland). The supernatants were harvested five days after transfection for analysis by ELISA and concentrated about 10 times with GE Healthcare Vivaspin^TM^ 2 concentrator (30,000 Da cut-off, ThermoFischer Scientific) for interaction with low-density PBMCs.

### 2.5. Detection of VB by ELISA

Costar 96-well plates were coated with anti-CH3 at 4 °C overnight (clone 57H, 1 µg/mL). After blocking plates with 1% BSA (Sigma-Aldrich, Merck KGaA, Darmstadt, Germany) in PBS, supernatants from transfected 293T cells were added as three-fold serial dilutions and incubated for two hours at 37 °C. The plates were next washed and incubated with biot-anti-CH3 (clone HP-6017) diluted 1:1000 in PBS with 1% BSA for 1 h at 4 °C. After washing, horseradish peroxidase (HRP)-conjugated streptavidin (Sigma-Aldrich, Merck KGaA) diluted 1:1000 in PBS with 1% BSA was added and plates were incubated 30 min at 4 °C. The plates were subsequently washed and developed with a tetramethylbenzidine (TMB) substrate reagent (Thermo-Scientific Scientific). OD was measured at 450 nm using a TECAN microplate reader (Tecan Group Ltd., Männedorf, Switzerland).

### 2.6. Interaction of VB with Pig Low-Density Peripheral Blood Mononuclear Cells (PBMCs)

Peripheral mononuclear blood cells were enriched in low-density cells using an OptiPrep (Sigma-Aldrich, Merck KGa) gradient as we previously described [42]. The cells (3 × 106 per point) were saturated with 5% pig serum in PBS and incubated with 10 times concentrated VB-containing supernatants for 30 min at 4 °C. The bound VB were revealed with anti-CH3 mAb followed by an A488-GAM IgM. As a negative control, we used concentrated supernatants from 293T cells transfected with a control plasmid (pcDNA3.1-Ctrl). Cells were co-labelled with anti-MHC class II mAb followed by A647-GAM IgG2a and anti-CD172A mAb followed by APC-cy7-GAM IgG2b and the specificity of labelling was assessed with appropriate isotype controls (fluorescence minus one). Dead cells were excluded with DAPI staining. The cells were analyzed using a Fortessa cytometer (Becton Dickinson & Company, Franklin Lakes, NJ, USA) and the FlowJo (v10.1) software (FlowJo, Becton Dickinson & Company).

### 2.7. Cationic Poly-Lactide Co-Glycolide Acid (PLGA) Nanoparticles (NPs)

The PLGA RG502H polymer (Sigma-Aldrich, Merck) was dissolved in methylene chloride as a 5% (*w/v*) solution and 1 mL was emulsified in with 0.1 mL of HEPES at 20 mM pH 7.4 at high speed using a probe sonicator. The primary emulsion was then added to 5 mL of distilled water containing cetyltrimethylammonium bromide (CTAB, Sigma-Aldrich, Merck) (0.5%, *w/v*). The resulting PLGA-CTAB nanoparticles (NPs) were washed once in buffer by centrifugation at 10,000× *g*. The PLGA nanoparticles were provided lyophilized and mixed were re-suspended with injectable NaCl 30 min before use. The DNA suspension was mixed with the cationic PLGA NPs at a 1:20 ratio (NPs:DNA, µg:µg). This formulation avoided precipitation, and the size of the NPs was measured to be between 120 and 140 nm with a poly-dispersion index around 0.4. Besides, a drop of the zeta potential monitored observed after the addition of DNA (from +15/+30 mV to −30/−50 mV) was a first indication that the negatively charged pDNA was adsorbed to the surface of the cationic NPs. This was confirmed by the measurement of free DNA (non-adsorbed to the NPs) in the formulation supernatant (after centrifugation). According to our calculation, the DNA/NPs loading ratio was between 15% and 20%.

### 2.8. Immunization of Pigs

All experiments were conducted in accordance with the EU guidelines and the French regulations (DIRECTIVE 2010/63/EU, 2010; Code rural, 2018; Décret n2013-118, 2013). The experimental procedures were evaluated and approved by the Ministry of Higher Education and Research (Notifications: APAFIS#413-2015051418327338 v10 (03/07/2017)). Pig procedures were evaluated by the ethics committee of the Val de Loire (CEEA VdL, committee n19) and took place at INRA Experimental Infection Platform PFIE (Nouzilly, France). Pigs (one-month-old) were obtained from a PRRSV-free herd, further confirmed by a commercial ELISA. After a week of acclimation, pigs were split into eight groups including eight piglets per group, with an equal number of males and females per group, except the MLV-only group which contained nine pigs. The groups’ description and the immunization schedule are shown in Figure 2A,B. On Day 0 (D0), Large-White pigs were anesthetized (2% isofluorane) and DNA plasmids (400 µg each in 400 µL saline) formulated on cationic PLGA NPs (1:20, NPs:DNA) were injected intradermally in three spots in the inguinal zone followed by surface electroporation on the site of injection using the CUY 21 EDIT system (NEPA GENE Co. Ltd., Chiba, Japan). Six electric pulses at 670 V/cm were applied during 10 ms with 90 ms interval (our unpublished data, see companion manuscript [35]). One group of eight pigs was kept as non-vaccinated controls. Trolamine 0.6% (Biafine, Johnson & Johnson Santé Beauté, Issy-les-Moulineaux, France) was spread on the transfected zone right after administration. On D29, the DNA 3X groups received the same DNA vaccines as on D0. The MLV-only and DNA + MLV groups were moved to A-BSL2 facility and they received MLV-FL13b (10 ^5.5^ TCID_50_ per pig) intramuscularly in 2 mL PBS + Ca^2+^Mg^2+^. The last injection of DNA was done on D62 in the DNA 3X groups. The pigs of the DNA 3X groups were euthanized by an overdose of sodium pentothal on D103 and the pigs of the DNA + MLV groups were euthanized on D75.

### 2.9. Overlapping Peptides

Overlapping peptides (20-mers, offset 8) covering the NSP1β, RdRp, N and the GP4GP5M PRRSV antigens were synthetized by Mimotopes (Mimotopes Pty Ltd., Victoria, Australia, http://www.mimotopes.com). Upon receipt, the peptides were diluted in H20:acetonitrile (50:50 vol) at a 5 mg/mL concentration and grouped as pools of peptides not exceeding 50 peptides: pool N (15 peptides), pool GP (GP4GP5M chimera, 40 peptides), pool NSP1β (47 peptides) and 2 pools RdRp1 (peptide 1–40), and pool RdRp2 (peptide 41–80). A 20-mer peptide from the HIV polymerase was used as control.

### 2.10. ELISPOTS on PBMCs from the DNA 3X and DNA + MLV Groups

PBMCs were collected on D75 (DNA 3X groups) and D42 (DNA + MLV) on three 8 mL Vacutainer^®^ CPT™ (BD-Bioscience) by centrifugation at 1800× *g* for 35 min at 25 °C. PBMCs were washed with PBS + 1.3 mM citrate and re-suspended in X-VIVO-20 medium (Ozyme, Saint-Cyr-l’Ecole, France) + 50% FCS + 1.3 mM citrate and rested overnight. PBMCs were resuspended in X-VIVO-20 medium supplemented with 2% FCS, 100 U/mL penicillin and 1 µg/mL streptomycin and counted for live cells. IFNγ-secreting T-cells were detected using PVDF membrane-bottomed 96-well plates (MultiScreen^®^_HTS_, Millipore, Merck KGaA) coated with 15 µg/mL anti-porcine IFNγ (capture mAb) in PBS. PBMCs (2 × 10^5^) were plated per well and were stimulated with the different pools of overlapping peptides described above at a 10 µg/mL final concentration for 18 h, in triplicates. A HIV polymerase-derived peptide (Mimotopes Pty Ltd.) and ConA at 25 µg/mL were used as controls. After 18 h, the IFNγ-secreting cells were revealed by sequential incubations with 0.5 µg/mL biotinylated anti-IFNγ followed by 0.5 µg/mL alkaline phosphatase conjugated-streptavidin (MabTech AB, Nacka Strand, Sweden) and 1-Step^TM^ BCIP/NBT reagent (Millipore, Merck KGaA). The spots were enumerated using the iSPOT reader from AID Autoimmun Diagnostica GmbH (Straßberg, Germany). Positive wells were considered if the mean spot numbers in the stimulated conditions were significantly superior to the spot numbers in the control peptide conditions (*p* < 0.05, paired *t*-test) and if >55 spots. The mean number of spots from stimulated minus control peptide wells was calculated.

### 2.11. Collection of Serum and Nasal Fluid and Evaluation of Their Content in Anti-N IgG by ELISA

Sera were collected on D0, D29, D62 and D103 in the DNA 3X groups and on D0, D29, D36, D42, D50 and D75 in the DNA + MLV groups. Nasal swabs were collected from the two nostrils in PBS + cOmplete Protease Inhibitor Cocktail (Roche, Sigma-Aldrich, Merck) on D0 and D75 in the DNA + MLV groups. Sera and swab collections were kept at −20 °C. The sera and swab extracts were assayed using the Ingezim PRRS 2.0 kit (Ingenasa, Eurofins-technologies, Bruxelles, Belgium) at a 1:40 and 1:2 dilution, respectively. The Ingezim PRRSV 2.0 kit is an PRRSV-1 and -2 nucleocapsid-based indirect ELISA, and present good sensitivity and specificity when compared to other commercial ELISA tests [43]. S/P ratio were calculated as follows: (OD sample minus OD negative control):(OD positive control minus OD negative control) with negative control being the mean OD value of the experiment’s negative control pig sera.

### 2.12. Viral Detection by Specific qRT-PCR

For quantification of viral RNA copies in sera collected on D29, D36, D42 and D50, viral RNA was first extracted from 100 µL serum using NucleoSpin^®^ RNA Virus kit from Nagel-Macherey. A one-step TaqMan qRT-PCR was performed using the iTaq ^TM^ universal probes One-Step kit (BIO-RAD, Hercules, CA, USA) with specific FL13 primers located in ORF1: forward primer 5’-TGTTTCCCCACAGATGTTTCG-3’, reverse primer 5’-CCAGGATTTTGAGGCTTTTCC-3’ and fluorescent probe FAM 5’-CCCCGAGTCAGTATC-3’ TAMRA (Sigma-Aldrich, Merck). The FL13b RNA standard curve was made with 10-fold dilutions from 10^1^ to 10^5^ TCID_50_/mL, each spiked in control pig serum and extracted with the NucleoSpin^®^ RNA Virus kit. The qRT-PCR was performed with 2 µL of sample elution in 10 µL final mix and the cycling involved the following steps: reverse transcription at 50 °C for 10 min, denaturation at 95 °C for 1 min, amplification 40 cycles at 95 °C for 3 s and 60 °C for 30 s. TaqMan run of experimental samples contained 2 replicates, mock pig serum RNA and H20. The TCID_50_ equivalent per mL (TCID_50_ eq/mL) serum was calculated from the FL13 standard curve, and the limit of detection was estimated to be 100 TCID_50_ eq/mL. The reactions were carried out in a CFX Connect^TM^ light cycler (BIO-RAD).

### 2.13. Statistical and Correlation Analysis

Data were analyzed with the GraphPad Prism 6.0 software (San Diego, CA, USA). The unpaired non-parametric two-way Mann–Whitney test was used to compare the ELISPOT results across groups and the IgG in sera and swab extracts.

## 3. Results

### 3.1. Production and Biochemical Characterization of Vaccibodies (VB) for Targeting PRRSV-Ags to Pig XCR1 and CD11c

We selected sequences from six PRRSV AG derived from the FL13 isolate (PRRSV-1, subtype 1) to be expressed by DNA vectors in a native form (untargeted, UT) or in vaccibody platforms in order to target these AG to APCs with the goal to induce high and broad T-cell responses (Figure 1A,B). The antigenic sequences have to be devoid of retention membrane motives for secretion as vaccibodies and for targeting APCs. NSP1β and RdRp do not contain predicted retention membrane motives as determined with the Annie prediction tool [41] and contain conserved T-cell epitopes across PRRSV-1 viruses [15]. A GP4GP5M fusion was built as a chimera that has been previously used to elicit high CD4^+^ T-cell responses with DC-SIGN targeting using PRRSV-2 sequences [39]. The M protein is also a well-conserved T-cell antigen across strains [14,15] and is included in the chimera. The N antigen is an immunodominant B-cell AG and it can be used to probe for the effects of targeting on antibody responses. VBs were constructed with these six PRRSV AG sequences, and with pig XCL1 and anti-pig CD11c scFv derived from the 3A8 clone [29] as targeting sequences. These pVB-PRRSV-AG constructs were initially transfected into 293T cells. The VBs with RdRp could not be detected in cell lysates, and they were thus not pursued. As XCL1-NSP1β was detected in cell lysates but not in the supernatants, NSP1β was split into two parts (NSP1β1 from AA 1 to 202 and NSP1β2 from AA 182 to 383). As shown in Figure 3, VBs were detected with an anti-CH3 ELISA in the supernatants of 293T cells transfected with pscCD11c-N, pscCD11c-GP4GP5M, pXCL1-N, pXCL1-GP4GP5M, pXCL1-NSP1β1, and pXCL1-NSP1β2 and they appeared to be expressed at variable levels.

### 3.2. Interaction of VB with Pig Cells

The supernatants from 293T cells transfected with the pVB-PRRSV-AG plasmids were assayed for their capacity to bind pig myeloid cells and cDC1, using PMBCs (Figure 4A) and low-density PMBCs (Figure 4B) respectively. Cells were incubated with the supernatants (10 times concentrated) and VBs were detected with the anti-CH3 mAb. ScCD11c-N and scCD11c-GP4GP5M bound MHC class II ^+^ cells (Figure 4A) and, as shown for scCD11c-N, they bound all CD172A^+^ MHC class II ^+^ cells, indicating that the scCD11c-VB interact with pig myeloid cells, as we reported before for the parental anti-CD11c IgG [29]. A fraction of the MHC class II^+^ CD172A^-^ cells was also labeled, and may correspond to cDC1 or CD3^+^ cells, as reported before [29]. XCL1-N and XCL1-GP4GP5M were detected to bind MHC class II^high^ FSC^high^ CD172^low^ low-density PBMCs, corresponding to cDC1 [29,42] (Figure 4B). This experiment confirmed that VB-PRRSV-AG targeted to CD11c and XCR1 are able to interact with pig myeloid antigen-presenting cells and cDC1 cells, respectively.

### 3.3. DNA Vectors Encoding PRRSV-AGs in Native Forms or Targeted to Pig APC Induce IFNγ T-Cell Responses in Pigs (DNA 3X Groups)

Pigs (one-month-old, eight per group) were immunized with DNA plasmids encoding VB-PRRSV-AGs or UT PRRSV-AGs. The plasmids received by the different groups are listed in Figure 2A (XCL1, scCD11c, and UT groups) and the immunization schedule and the sampling timeline is shown in Figure 2B. The plasmids (400 µg each) were combined to cationic PLGA-NPs at a 20:1 DNA:NPs ratio (µg:µg) with the goal to promote DNA stability in extracellular fluid and during the intracellular trafficking to the nucleus [34]. Plasmids associated with cationic PLGA-NPs were administered intradermally with surface electroporation (EP). This combination was shown to be highly efficient at inducing immunogenicity [34] and in a parallel study in our laboratory (unpublished data, companion manuscript [35]). Three administrations were performed at a month interval (DNA 3X groups). PBMCs were collected 13 days after the third boost and processed to IFNγ ELISPOT analysis using re-stimulating overlapping peptide pools covering the different PRRSV antigenic sequences. The response against RdRp was evaluated with two peptide pools (N and C terminal), in order not to exceed 40 peptides per pool and avoid toxicity. Figure 5 shows that the different plasmid vaccines induced IFNγ responses, with the GP4GP5M peptide pool revealing the lowest responses and the pRdRP pools showing the highest response, which reached statistical significance versus the control group. The number of recognized peptide pools was calculated for each pig. In the immunized groups, the majority of pigs responded to at least one peptide pool, although few pigs in each group displayed no reactivity to the tested peptide pools (Figure 5F, left panel). No benefit of targeting N, GP4GP5M and NSP1β to XCR1 or of targeting N and GP4GP5M to CD11c was obtained.

### 3.4. A DNA-MLV Prime–Boost (DNA + MLV Groups) Broadens the IFNγ T-Cell Response as Compared to MLV-Only, Induces Higher T-Cell Responses as Compared to DNA 3X, and Does not Show a Benefit of APC-Targeting on the IFNγ T-Cell Response

We hypothesized that a DNA-MLV prime–boost strategy could be able to enhance the T-cell response induced by MLV and might be superior to a DNA-stand-alone strategy. We therefore compared the efficacy of this strategy to the DNA 3X’s one. We immunized one-month-old pigs with the DNA plasmids on the same day, in the same housing and with the same plasmids as for the DNA 3X groups (Figure 2A). On Day 29, the DNA + MLV pigs and MLV-only immunized pigs received MLV-FL13b (10^5.5^ TCID_50_ per pig) by the intramuscular route. PBMCs were collected on D13 after the MLV boost and processed similarly as described above for the DNA 3X groups. For all peptide pools except GP4GP5M, the MLV-only and DNA + MLV groups showed statistically significant IFNγ T-cell response above the unimmunized control group (Figure 5A,C–E). Pigs of the DNA + MLV groups showed higher responses than pigs immunized with MLV-only but due to the heterogeneity of response, the difference did not reach statistical significance (Figure 5A–E). The MLV-only vaccinated pigs developed similar levels of IFNγ T-cell response as the vaccinated DNA 3X pigs. Whereas two pigs of the MLV-only group did not present detectable T-cell responses, all pigs of the DNA + MLV groups responded to at least one peptide pool (Figure 5F). These DNA-MLV prime–boost groups responded to more peptide pools than did the MLV-only and their respective DNA 3X ones, and there were no non-responders (Figure 5F, *p* < 0.05). Therefore, altogether, the results indicate that the DNA-MLV prime–boost strategy induces higher and broader IFNγ T-cell response as compared to a DNA-stand-alone strategy or as MLV-only.

### 3.5. A DNA-MLV Prime–Boost Potentiates the Anti-N Igg Response as Compared to an MLV-Only Vaccine or to A DNA-Stand-Alone Regimen and Shows a Benefit of XCR1 and CD11c-Targeting 

N is an immuno-dominant B-cell antigen, which was successfully targeted to XCR1 and CD11c with VBs. The IgG anti-N response was detected in the sera with a robust commercial ELISA kit [43,44]. In the DNA 3X groups, a low anti-N IgG response was detected after the first injection and it gradually increased with the boosting, except in the CD11c group where the third boost had no effect. The mean S/P ratio in the XCL1, scCD11c and UT group reached the maximal values 0.97 ± 0.41, 0.19 ± 0.07 and 0.96 ± 0.66, respectively, on D103 (Figure 6A). The anti-N IgG response was strongly enhanced in the DNA + MLV groups as compared to in the MLV-only, as soon as on D36 (i.e., seven days after the MLV boost). The mean S/P ratio at D36 reached 1.14 ± 0.43 in the XCL1 group and 1.21 ± 0.92 in the CD11c group, which are higher values than in the DNA 3X at any time point (Figure 6B). The anti-N IgG response started to be detectable at D42 in the MLV-only group (i.e., 13 days after the boost, mean S/P ratio = 0.12 ± 0.08). At D50, the mean S/P ratio reached a maximum in the XCL1 and scCD11c prime–boost groups (4.64 ± 0.57 and 4.24 ± 0.79), whereas the value was much lower, i.e., 0.77 ± 0.34, in the MLV-only group. Interestingly, the S/P ratios were higher in the XCL1 and scCD11c than in the UT prime–boost group at D50 (*p* < 0.05), as well as at D42 and D75 in the XCL1 group (*p* = 0.06 and *p* < 0.05, respectively). As the MLV + DNA pigs were culled on D75, the duration of the anti-N IgG response in the three groups could not be evaluated beyond that time point. Overall, this analysis shows that the DNA-MLV prime–boost strongly accelerated the anti-N IgG response induced by MLV and the magnitude of the anti-N IgG response was much higher with this regimen than with DNA 3X or MLV alone, in the time frame of the experiment. Furthermore, APC-targeting and especially XCR1 targeting potently enhanced the anti-N IgG response.

### 3.6. Anti-N IgG Response in Nasal Secretions

Thus far, the DNA-MLV prime–boost strategy appears as the most potent one to induce broad IFNγ T-cell response and high/rapid antibody response against PRRSV. As the achieved antibody magnitude is striking, this DNA-MLV prime–boost strategy might be used to protect the mucosa, the portal of entry of PRRSV. Although anti-N Abs are not protective, they can be considered as representative to evaluate the potency of the DNA-MLV for induction of mucosal antibodies. Besides, the GP4GP5M chimera does not induce measurable neutralizing antibodies, as published by others [39,45]. No anti-N IgA were detectable in nasal swabs collected at D75 but anti-N IgG were detected with the commercial ELISA. Figure 6C shows that, at D75, the level of anti-N IgG tended to reach the highest levels in the XCL1 group as compared to in the MLV-only group. This finding indicates that the prime–boost strategy, especially with XCR1-targeting of vaccine antigens, promote PRRSV-specific Abs location in mucosa.

### 3.7. A DNA Prime Effect on MLV Replication

MLV replication can lead to reversion to pathogenicity [6]. We therefore evaluated whether the DNA prime had any impact on MLV replication. MLV-FL13b was detected by TaqMan qRT-PCR in the pig sera of the DNA + MLV groups on D29, D36, D42 and D50 (Figure 7 and Appendix A). Low amounts of MLV-FL13b, showing variable patterns over time between pigs, were detected (below 10^3^ TCID_50_ eq/mL in most instances, Appendix A). Figure 7 illustrates that viral RNA amounts over time were similar across groups. Therefore, although the DNA prime stimulated immunity, it was not sufficient to reduce MLV replication.

## 4. Discussion

Our results show that DNA vectors expressing a combination of conserved PRRSV antigens inoculated in pig dermis with cationic PLGA NPs and surface electroporation induces moderate but significant IFNγ T-cell and antibody responses. Interestingly, we showed that the DNA priming with these vectors enhanced the breadth of the IFNγ T-cell response and the level of anti-N IgG induced by MLV, in the time frame of our experiment. The antibody response induced with MLV raised slowly whereas the antibody response induced with DNA + MLV rapidly reaches very high levels and started to decline at D75. Given the constraints of the large animal experiment scheduling, pigs were culled at D75, which prevented us to determine the subsequent evolution of the antibody levels in the different groups. In any event in this time frame, we observed a benefit of targeting a PRRSV antigen to APCs, and especially to XCR1, for inducing elevated amounts of antibodies, in both serum and secretions.

Our initial objective was to develop a PRRSV T-cell vaccine, and we hoped to improve the T-cell responses with APC-targeting. We did not achieve this goal, and we rather obtained higher antibody responses with this approach and only in the context of the prime–boost strategy. Of note, we only monitored the IFNγ T-cell responses, and no other arms of the T-cell response, which could behave differently than the IFNγ T-cell response upon APC-targeting. Several published works emphasize that the outcomes of DC targeting remain difficult to predict, especially in non-model species [46,47,48]. The T-cell antigen epitope content probably strongly impacts on the efficacy of DC-targeting for inducing IFNγ T-cell response: indeed, contrasting results were obtained with DEC205 targeting of two different plasmodium vivax antigens, with benefit on the IFNγ T-cell response for one and no effect for the other [49]. Some antigens may include high amounts or potent regulatory T-cell epitopes, affecting the outcome of DC-targeting, with variations between individuals also depending on their genetic make-up and their previous immune experience [50,51]. In this study, only the N antigen and the GP4GP5M chimera could be efficiently expressed in the context of targeted VBs; two parts of NSP1β could be expressed with XCL1-VB but the levels of expression were low in 293T cells, and VB including RdRp, which revealed to be the most potent T-cell antigen in our ELISPOT experiments, were not expressed in cells. A previous work with a GP4GP5M chimera indicated that IFNγ T-cell response could be increased with DC-SIGN targeting [39], but these results were not confirmed by a second study using a different adjuvant [45]. We observed that this chimera is inefficient at inducing IFNγ T-cell response in our setting, whether targeted or not. Finally, NSP5 would have been a very interesting T-cell antigen to be used in a DNA-MLV prime–boost strategy, as NSP5 is the most conserved and dominantly recognized T-cell antigen in pigs infected with subtype 1 and 3 viruses [14]. However, NSP5 includes many predicted transmembrane domains and was therefore not suitable for expression with VB.

Our delivery method with cationic PLGA NPs and surface EP may interfere, positively or negatively, on the efficacy of the DNA-MLV prime–boost strategy on the immune responses, and possibly on the effect of APC-targeting. Indeed, the effects of cationic PLGA NPs associated to EP, previously shown to be beneficial on DNA vaccine immunogenicity [34], may act by improving the sensing of the DNA, by stabilizing the DNA, by promoting its intracellular transport towards the nucleus, and/or by favoring the direct transfection of APC in skin, especially after the electric pulse. In addition, we showed that the PLGA-NPs with the EP delivery method has an inflammatory and adjuvant influence (our unpublished data, see the companion manuscript [35]). This adjuvant effect may be necessary to the efficacy of the DNA-MLV prime–boost strategy on some arms of the immune responses. Alternatively, on the opposite, cationic PLGA NPs associated to EP may have led to a maximal IFNγ T-cell responses which cannot be exceeded with DC-targeting of the DNA encoded antigen, as previously shown with gradation of plasmid doses encoding for DEC205-targeted antigens in the mouse [22]. These hypotheses are evaluated in a second submitted manuscript, where PLGA NPs were not used.

Our DNA constructs used as a stand-alone strategy induced anti-N responses to moderate but substantial levels after three injections. However, no boosting effect was obtained with the third boost using the scCD11c-N VB vector. The inefficacy of homologous boosting has been previously reported when CLEC9A was targeted with a xenogeneic antibody [52]. Indeed, the xenogeneic antibody sequences, here represented by mouse scFv, are presumably intrinsically immunogenic in the pig and the anti-scFv response mounted by the pig could block the boosting effect, as demonstrated in the CLEC9A-targeting study [52]. Conversely, pXCL1-N VB, which contain syngeneic pig XCL1 sequences, are not unfavorable in successive boosting. In the CLEC9A-targeting experiment, the authors showed that DC-targeting was particularly efficient when used with a heterologous boost [52]. Indeed, in our study, the DNA-MLV prime–boost strategy potently favored the antibody response, and the response was promoted by XCR1 and CD11c targeting, and even in mucosa. XCR1-targeting appears to be particularly suitable to promote antibody response both in mice [24,53] and in pigs as previously shown by us [54]. It has also been proposed that XCR1-targeting with ligands that induce a slow internalization kinetics of XCR1 [53] allows a prolonged antigenic interaction, leading to efficient activation of the BCR. However, a slow internalization kinetics of XCR1 would not be favorable to CD8^+^ T-cell activation. It is possible that, in the setting that we used in pigs, the binding properties of XCL1 VB favors more the antibody response than the CD8^+^ T and CD4^+^ Th1 cell response. As said in the results, the anti-N IgG response was monitored as an “emblematic” antibody response. However, protective antigen determinants, yet to be identified, could be used in VB in a DNA-MLV prime–boost strategy to elicit strong neutralizing antibodies, and ideally broadly neutralizing ones, which exist in some PRRSV-infected pigs [55].

The DNA-MLV prime–boost strategy, independently of the APC-targeting, promoted the IFNγ T-cell response. Of note, with the DNA-only regimen despite the use of three injections, some of the pigs did not respond to the tested peptide pools, whereas, with the DNA-MLV regimen, all pigs responded at least to one peptide pool. Non-responders in IFNγ ELISPOTS are frequently encountered upon pig immunization with PRRSV vaccines [15,56], therefore the combination of DNA with MLV, through two complementary antigen-presentation processes, appears to promote T-cell epitope recognition and avoid non-responders. Another group have previously showed the benefit of a DNA-MLV prime–boost using DNA vectors encoding a truncated N devoid of immunomodulatory property which reduced the N-specific T regulatory response induced by MLV [57]. In our experiment, we could not detect IL-10 production in the supernatants of re-stimulated cells and we therefore could check if our multiple PRRSV antigens also had the interesting property to limit the immune inhibition induced by PRRSV MLV. IFNγ ELISPOT assay is often used to probe T-cell responses in pigs, however this assay does not capture the complex nature of the memory T-cell responses, and more selective and precise arms of this response may correlate better with protection. Indeed, the IFNγ CD8^+^ T-cell or cytolytic responses may be more pertinent to protection than IFNγ CD4^+^ T-cell responses [58] as well as the orientation of the T-helper cell polarization or its poly-functionality. Furthermore, some T-cell epitopes may be associated to protection and not others. In this work, we also used only a selection of PRRSV-AGs, which should be extended to others, such as NSP5 [14]. Although improved IFNγ T-cell responses were achieved with our DNA-MLV prime–boost strategy, a single administration (priming) with the chosen PRRRV-AGs appeared not sufficient to control the MLV replication. Addition of other T-cell PRRRV-AGs may improve this control and therefore the safety of MLV, as well as the aimed heterologous protection.

## 5. Conclusions

Our work shows that the DNA-MLV prime–boost strategy broadens the T-cell response and increases the antibody response over DNA-only and MLV-only regimens, and it can now be considered for the rational design of broadly efficacious vaccines against PRRSV. Its protective efficacy needs to be evaluated in a heterologous challenge, also with adding other T antigens such as NSP5. In practice, provided that cost-effectiveness is favorable to field application, the DNA-MLV strategy could also be used in gilts in acclimation or in neonatal piglets under colostrum immunity. Indeed, DNA vaccination appears poorly sensitive to the effects of maternal antibodies, differently from MLV [59,60], and, after weaning, the DNA-primed piglets could be vaccinated with MLV and they might develop rapid and broad immunity upon transfer to fattening units. Studies of compared efficacy, economic assessments and effectiveness modeling will have to be undertaken in order to bring such PRRSV DNA vaccine component to the field.

## Figures and Tables

**Figure 1 viruses-11-00551-f001:**
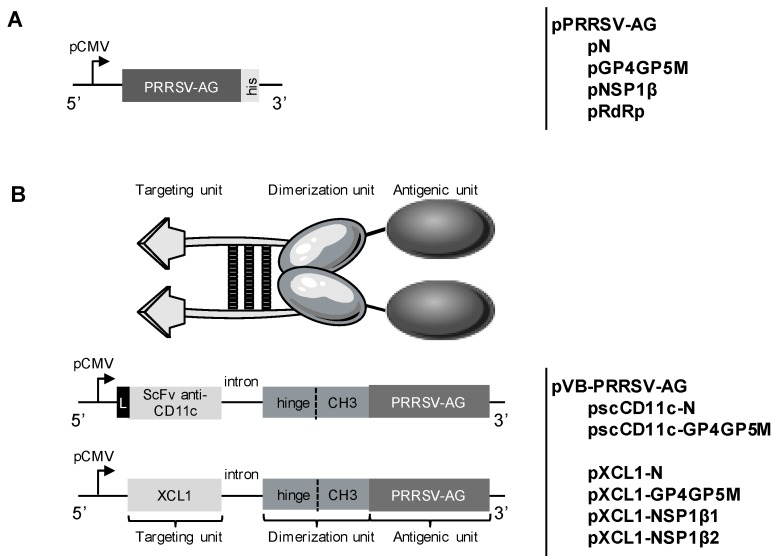
DNA constructs for expression of PRRSV-AGs in native forms or in VBs targeting XCR1 and CD11c. (**A**) Four PRRSV AG constructs encompassing six PRRSV AG (FL13 isolate) were codon-optimized, terminated by a 6× His tag and cloned under the CMV promoter. The four constructs are pN, pGP4GP5M chimera, pNSP1β and pRdRp. (**B**) The VB-PRRSV-AGs include a targeting unit, a dimerization domain derived from the hinge and CH3 domains of human IgG3, and the four PRRSV-AGs described in (**A**). The VB sequences were also cloned under the CMV promoter. Two constructs, pscCD11c-N and pscCD11c-GP4GP5M, include an anti-pig CD11c scFv sequence for targeting the N and GP4GP5M chimera, respectively; the added exogenous murine Ig leader sequence is represented at the 5’ end by a black box. Four constructs, pXCL1-N, pXCL1-GP4GP5M, pXCL1-NSP1β1 and pXCL1-NSP1β2 target the N antigen, the GP4GP5M chimera, the AA1-202 NSP1β1 fragment and the AA182-383 NSP1β2 fragment, respectively.

**Figure 2 viruses-11-00551-f002:**
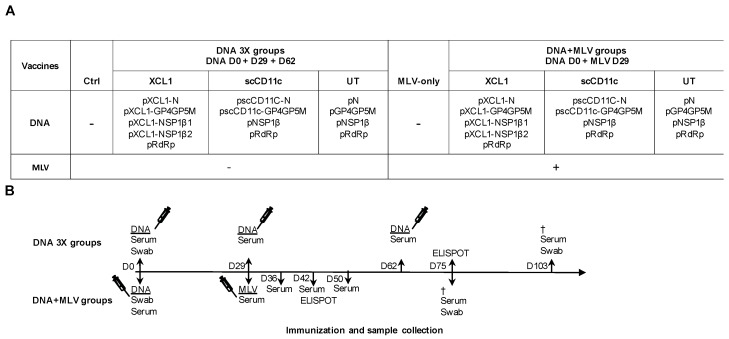
Immunization protocol with DNA-only (DNA 3X) and with DNA-MLV prime–boost (DNA + MLV). (**A**) One-month-old pigs (eight per group) of the DNA 3X and DNA-MLV groups received 400 µg of the plasmids listed in the table, on the same day (D0). The plasmids were formulated with PLGA NPs and delivered intradermally using surface electroporation. The groups are named under the VB type used to target DC (XCL1 or scCD11c), the untargeted antigens (UT) or under controls (not immunized). Note that, in all groups, pRdRp used as RdRp could not be targeted, and in the scCD11c group, pNSP1β coding for UT NSP1β was used. On D29 and D62, the DNA 3X groups were boosted with the same DNA plasmids as on D0. On D29, the four DNA + MLV groups were injected with MLV-FL13b (10^5.5^ TCID_50_ per pig) intramuscularly. (**B**) The sampling and immunization schedules are reported on the time line: blood was collected on CPT tubes for ELISPOT assay on D42 and D75 (DNA group and DNA + MLV group, respectively, 13 days after boost). The DNA + MLV groups were euthanized on D75 and the DNA 3X on D103. The sera were collected on D0, D29, D62, and D103 in the DNA 3X groups and on D0, D29, D36, D42, D50 and D75 in the DNA + MLV groups. Nasal swabs were collected on D0 and D103 in the DNA groups and on D0 and D75 in the DNA + MLV group.

**Figure 3 viruses-11-00551-f003:**
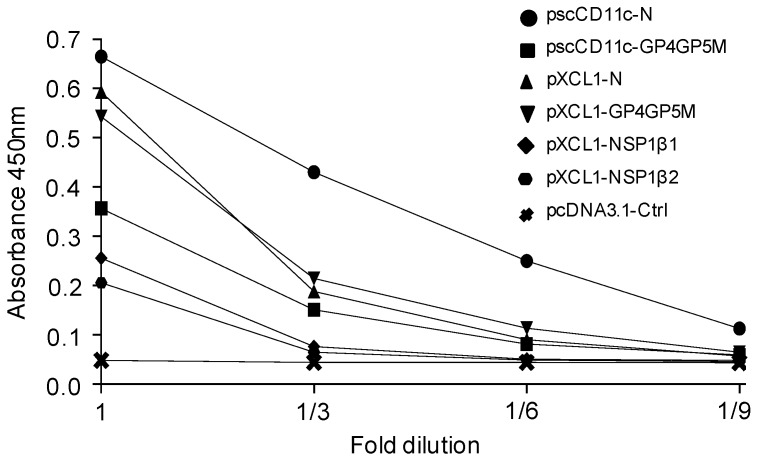
Expression of VB-PRRSV-AGs. The pscCD11c-N, pscCD11c-GP4GP5M, pXCL1-N, pXCL1-GP4GP5M, pXCL1-NSP1β1, pXCL1-NSP1β2 and pcDNA3.1-ctrl plasmid were transfected in 293T cells. The supernatants were harvested after five-day culture in serum-free medium and assayed in three-fold successive dilutions for VB detection in ELISA using anti-CH3 (capture) and biot-anti-CH3 (detection) antibodies (see material and methods). The absorbance at 450 nm is reported.

**Figure 4 viruses-11-00551-f004:**
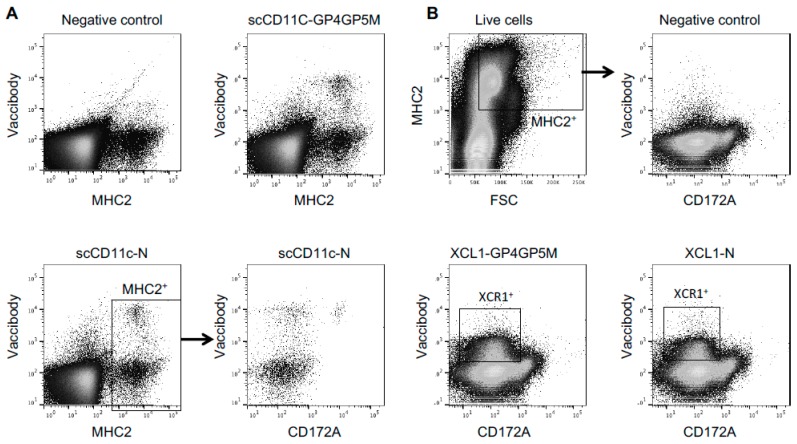
Binding of VB-PRRSV-AGs to whole pig PBMCs and low-density pig PBMCs. (**A**) PBMCs were incubated with about 10 times concentrated supernatants containing scCD11c-N and scCD11c-GP VBs as well as with a negative control concentrated supernatant (cells transfected with pcDNA3.1-ctrl), and the VBs were revealed with anti-CH3 mAb followed by A488-GAM IgM. The cells were co-stained for detection of pig MHC class II (A647) and CD172A (APC-cy7). Dead cells were excluded with DAPI. (**B**) Low-density PBMCs were incubated with about 10 times concentrated supernatants containing XCL1-N, XCL1-GP4GP5M, and with the negative control concentrated supernatant, followed by the same steps as in (**A**).

**Figure 5 viruses-11-00551-f005:**
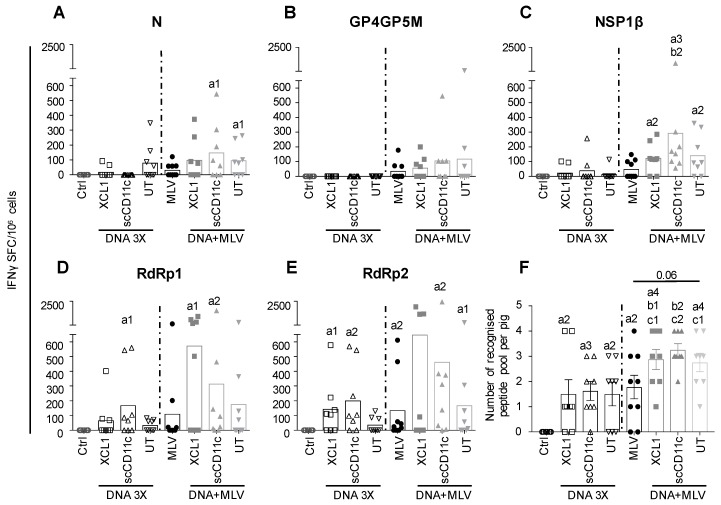
T-cell responses induced by DNA 3X and DNA + MLV regimens. PBMCs were collected 13 days after the last boost in the DNA 3X (D75) and DNA + MLV (D42) groups and they were analyzed for IFNγ T-cell response upon re-stimulation with overlapping peptide pools (10 µg/mL) in ELISPOT assays. (**A**–**E**) The responses to the N, GP4GP5M, NSP1β, RdRp1 (peptide 1–40) and RdRp2 (peptide 41–80) peptide pools are shown. An irrelevant peptide (10 µg/mL) was used as control. The mean number of spots from triplicates of stimulated minus control wells are shown (see Section 2). The mean number of spots across pigs per group is shown as a box. The plasmids used to immunize each group are listed Figure 2. (**F**) The number of recognized peptide pools per pig is shown for each group. Statistically significant differences between two groups were calculated using the Mann–Whitney non-parametric test. The *p* value is indicated when close to significance. The letter a indicates significant differences with the non-vaccinated group, b with the MLV-only group, and c with the respective DNA 3X group. Number 1 corresponds to *p* < 0.05, 2 to *p* < 0.01, 3 to *p* < 0.001, and 4 to *p* < 0.0001.

**Figure 6 viruses-11-00551-f006:**
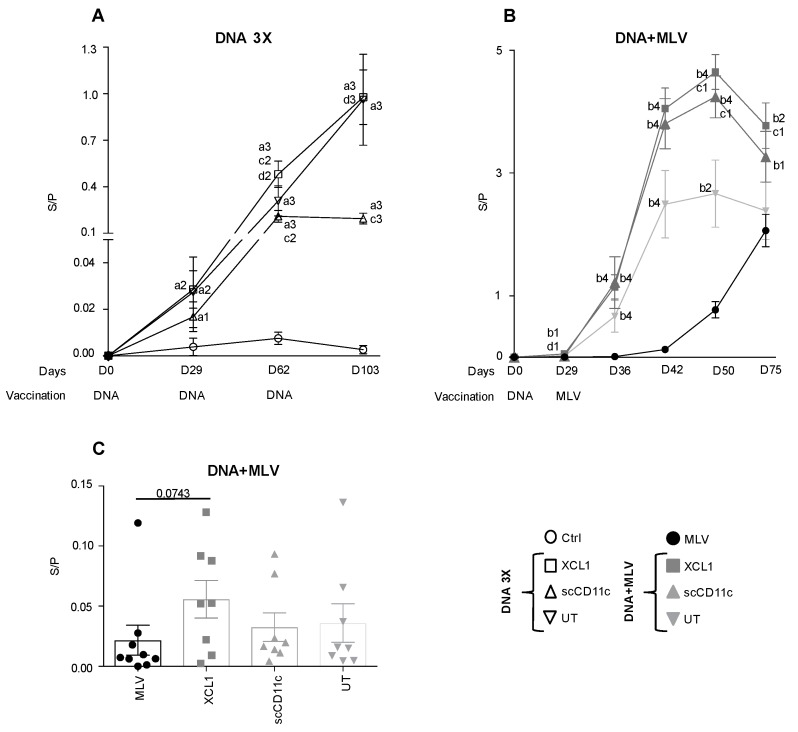
Anti-N IgG responses in serum and nasal secretions induced by DNA 3X and DNA + MLV regimens. (**A**,**B**) Sera from the DNA 3X groups and from the DNA + MLV groups were collected on the indicated days and analyzed at a 1:40 dilution for anti-N IgG detection with the Ingezim PRRS 2.0 kit. The S/P ratios are reported as means ± sem per group. The vaccine dates and types are shown. (**C**) Nasal secretions were collected on D0 and D75 in the DNA + MLV groups. The secretions were analyzed at a 1:2 dilution for anti-N IgG detection with the Ingezim PRRS 2.0 kit. The S/P ratios for each pig per groups are shown and the mean of the group (box) ± sem (error bars) is shown. Statistically significant differences between two groups were calculated at each time point using the Mann–Whitney non-parametric test. The letter a indicates significant differences with the non-vaccinated group, b with the MLV-only group, and c with the UT groups. Number 1 corresponds to *p* < 0.05, 2 to *p* < 0.01, 3 to *p* < 0.001, and 4 to *p* < 0.0001. The *p*-value is indicated when close to significance.

**Figure 7 viruses-11-00551-f007:**
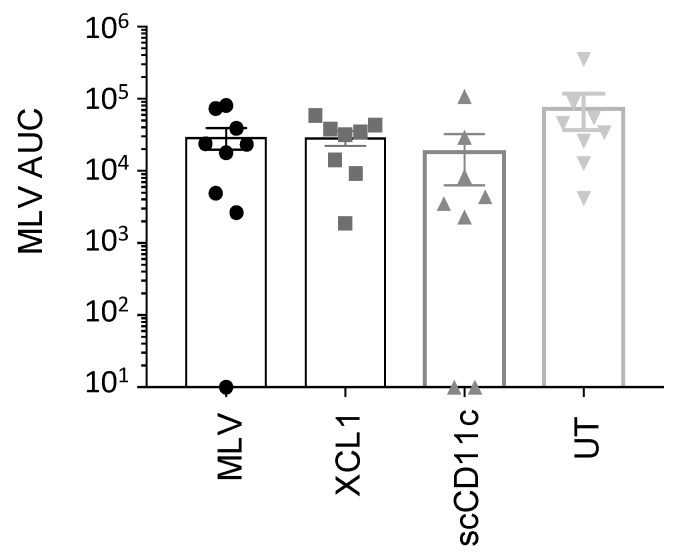
Detection of MLV-FL13b RNA in the sera of immunized pigs. MLV-FL13b was detected in sera with specific qRT-PCR on D29, D36, D42, and D50. TCID_50_/mL equivalence was calculated with a viral standard curve at each time point (see Material and Methods) and the area under the curve (AUC) was extrapolated for each pig. The mean (box) ± sem is shown per group. The symbols are the same as in Figure 6.

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
