# Peer review of "A DNA-Modified Live Vaccine Prime–Boost Strategy Broadens the T-Cell Response and Enhances the Antibody Response against the Porcine Reproductive and Respiratory Syndrome Virus"

_viruses, 2019, doi:10.3390/v11060551_

Reviewer 1 Report

The present paper deals with the use of a DNA vaccines including diferent T and ntigens of PRRSV1 expressed in native form or as vaccibodies. The study is an interesting piece of work well designed and performed. It presents some very relevant data (for exemple a confirmation of RdRP as a dominant T-cell antigen and the possibility of enhances very significantly antibody responses by targeting CD11c or XCR1). The paper is written clearly and in spite of the amount of information and the relative sophistication of the used techniques can be followed easily. The main criticisms are about the interpretation of the results. For exemple, the autors clearly indicated that only a fraction of the tested peptides was recognised. In fig 5 it seems that some pigs did not recognise any peptide pool, is that right? This is a major problem that arises every time a screening of PRRSV peptides is done and has serious practical implications when thinking of the use of such strategies in the field. This is not discussed. Similarly, as seen in fig 7, the use of the MLV in animals previously immunized with the DNA vaccines resulted in a viral replication similar to what happen in unvaccinated animals. This must be discussed in depth since indicates that the generated immunity was not able to reduce viral replication. Considering the predominance of RdRp in the ELISPOT responses, the question is obvious: are the IFN ELISPOT results a good indicator (correlate) for PRRSV immunity? Although used extensively in PRRSV research, the published results and the experience indicate that IFN ELISPOT results are tricky. In other words, higher magnitudes of IFN-secreting cells probably correlate with better protection but the pool of IFN-secreting cells most probably contain a mix of cells with specificities towards relevant (protective) and irrelevant (non-protective) epitopes. (by the way, RdRP probably plays a role in protection against PRRSV). The biological significance of the tests used must be discussed.

Finally, the conclusions must be totally revised. The vaccination approach seems unrealistic. In the present work 3X doses of DNA were needed. Under field conditions with endemic circulation of PRRS the approach would be unfeasible (consider that some MLV vaccines are licensed for use in 1 week old piglets). Maybe use for acclimation of sows would be more interesting, especially if neutralizinfg antibody responses could be enhanced.

Author Response

Reviewer 1- Italic = reviewer comment, normal black = our answer, in red = the lane where the modified text is included and the text itself.

The present paper deals with the use of a DNA vaccines including different T and antigens of PRRSV1 expressed in native form or as vaccibodies. The study is an interesting piece of work well designed and performed. It presents some very relevant data (for exemple a confirmation of RdRP as a dominant T-cell antigen and the possibility of enhances very significantly antibody responses by targeting CD11c or XCR1). The paper is written clearly and in spite of the amount of information and the relative sophistication of the used techniques can be followed easily. The main criticisms are about the interpretation of the results. For exemple, the autors clearly indicated that only a fraction of the tested peptides was recognised. In fig 5 it seems that some pigs did not recognise any peptide pool, is that right? This is a major problem that arises every time a screening of PRRSV peptides is done and has serious practical implications when thinking of the use of such strategies in the field. This is not discussed.

In Figure 5F, the number of peptide pools recognized per pig is represented. Indeed, as pointed out by Reviewer 1, in the DNA 3X groups as well as in the MLV-only group, some pigs in each group do not respond to any peptide pool. However, in the MLV+DNA groups, all pigs respond at least to one of the peptide pool. We now comment more clearly this finding:

L434 (results): In the immunized groups, the majority of pigs responded to at least one peptide pool, although few pigs in each group displayed no reactivity to the tested peptide pools (Figure 5F, left panel).

L455 (results): These DNA-MLV prime-boost groups responded to more peptide pools than did the MLV-only and their respective DNA 3X ones, and there are no non-responders (Figure 5F, p < 0.05).

L664 (discussion): Of note, with the DNA-only regimen despite the use of 3 injections, some of the pigs did not respond to the tested peptide pools, whereas with the DNA-MLV regimen, all pigs responded at least to one peptide pool. Non-responders in IFNg ELISPOTS are frequently encountered upon pig immunization with PRRSV vaccines [15, 56], therefore the combination of DNA with MLV, through two complementary antigen-presentation processes, appears to promote T-cell epitope recognition and avoid non-responders.

Similarly, as seen in fig 7, the use of the MLV in animals previously immunized with the DNA vaccines resulted in a viral replication similar to what happen in unvaccinated animals. This must be discussed in depth since indicates that the generated immunity was not able to reduce viral replication.

Considering the predominance of RdRp in the ELISPOT responses, the question is obvious: are the IFN ELISPOT results a good indicator (correlate) for PRRSV immunity? Although used extensively in PRRSV research, the published results and the experience indicate that IFN ELISPOT results are tricky. In other words, higher magnitudes of IFN-secreting cells probably correlate with better protection but the pool of IFN-secreting cells most probably contain a mix of cells with specificities towards relevant (protective) and irrelevant (non-protective) epitopes. (by the way, RdRP probably plays a role in protection against PRRSV). The biological significance of the tests used must be discussed.

The point is well taken and now discussed:

L675: IFNg ELISPOT assay is often used to probe T-cell responses in pigs, however this assay does not capture the complex nature of the memory T-cell responses, and more selective and precise arms of this response may correlate better with protection. Indeed, the IFNg CD8+ T-cell or cytolytic responses may be more pertinent to protection than IFNg CD4+ T cell responses [58] as well as the orientation of the T helper cell polarization or its poly-functionality. Furthermore, some T-cell epitopes may be associated to protection and not others. In this work, we also used only a selection of PRRSV-AGs, which should be extended to others, such as NSP5 [14]. Although improved IFNg T-cell responses were achieved with our DNA-MLV prime-boost strategy, a single administration (priming) with the chosen PRRRV-AGs appeared not sufficient to control the MLV replication. Addition of other T-cell PRRRV-AGs may improve this control and therefore the safety of MLV, as well as the aimed heterologous protection.

Finally, the conclusions must be totally revised. The vaccination approach seems unrealistic. In the present work 3X doses of DNA were needed. Under field conditions with endemic circulation of PRRS the approach would be unfeasible (consider that some MLV vaccines are licensed for use in 1 week old piglets). Maybe use for acclimation of sows would be more interesting, especially if neutralizinfg antibody responses could be enhanced.

We agree that the conclusion needs to be revised. Indeed, the DNA –only approach, with 3 injections, appears not suitable for future practical developments. We thus now modify the conclusion, promoting clearly the DNA-MLV strategy, and stating that a challenge has to be performed using our optimal conditions and completing the antigens with at least NSP5. We think that a DNA-prime has some meaning to be evaluated in neonates with maternal antibodies, as DNA vaccines appear less sensitive to maternal interference as MLV.

L689: Our work shows that the DNA-MLV prime-boost strategy broadens the T cell response and increases the antibody response over DNA-only and MLV-only regimens, and it can now be considered for the rational design of broadly efficacious vaccines against PRRSV. Its protective efficacy needs to be evaluated in a heterologous challenge, also with adding other T antigens such as NSP5. In practice, provided that cost-effectiveness is favorable to field application, the DNA-MLV strategy could also be used in gilts in acclimation or in neonatal piglets under colostrum immunity. Indeed, DNA vaccination appears poorly sensitive to the effects of maternal antibodies, at the difference with MLV [59, 60] and after weaning, the DNA-primed piglets could be vaccinated with MLV and they might develop rapid and broad immunity upon transfer to fattening units. Studies of compared efficacy, economic assessments and effectiveness modeling will have to be undertaken in order to bring such PRRSV DNA vaccine component to the field.

Reviewer 2 Report

The manuscript describes pig immune response towards new DNA-vaccines coupled with a MLV vaccine against PRRSV. The study is well-written and the experiments are appropriated to evaluate the immune response to the developed epitopes. Results reported here are in line with previous studies showing that DNA-MLV vaccination enhance immune responses to other viruses that require mucosal immunity.

I have, however, a comment regarding the conclusion of the manuscript. It seems precipitated to me to imply that this strategy is “a promising strategy to control PRRSV infection” as no challenge study has been conducted and more often than not great immune responses are not protective when animals are challenged. Another important consideration are the costs and feasibility to use such strategies in the field in the near future. A conclusion that highlights the main findings of the current study and indicate the need to further evaluate the proposed strategy would be more appropriated.

Minor comments:

Line 30: agents instead of agent

Line 55: the instead of he

Line 116-127: all antibodies described in this section are mAb, so it is unclear why the definition of mAb is given only in line 119 and line 120 states “the mAbs used in this study are…”

Fig. 2. The plus and minus signs are not very clear in the table and perhaps could be substituted

Line 281: extra-space after Australia

Line 323: extra ) after ml

Line 328: per instead of par

Figure 5. Add a description to the graphs depicted in 5F in the legend. Give meaning of 4 in the legend

Author Response

Reviewer 2 - Italic = reviewer comment, normal black = our answer, in red = the lane where the modified text is included and the text itself.

The manuscript describes pig immune response towards new DNA-vaccines coupled with a MLV vaccine against PRRSV. The study is well-written and the experiments are appropriated to evaluate the immune response to the developed epitopes. Results reported here are in line with previous studies showing that DNA-MLV vaccination enhance immune responses to other viruses that require mucosal immunity.

I have, however, a comment regarding the conclusion of the manuscript. It seems precipitated to me to imply that this strategy is “a promising strategy to control PRRSV infection” as no challenge study has been conducted and more often than not great immune responses are not protective when animals are challenged. Another important consideration are the costs and feasibility to use such strategies in the field in the near future. A conclusion that highlights the main findings of the current study and indicate the need to further evaluate the proposed strategy would be more appropriated.

We agree with this reviewer that the conclusion needs to be modified and should mention complementary studies, including socio-economics, which are mandatory before any field application can possibly be considered.

L678: Our work shows that the DNA-MLV prime-boost strategy broadens the T cell response and increases the antibody response over DNA-only and MLV-only regimens, and it can now be considered for the rational design of broadly efficacious vaccines against PRRSV. Its protective efficacy needs to be evaluated in a heterologous challenge, also with adding other T antigens such as NSP5. In practice, provided that cost-effectiveness is favorable to field application, the DNA-MLV strategy could also be used in gilts in acclimation or in neonatal piglets under colostrum immunity. Indeed, DNA vaccination appears poorly sensitive to the effects of maternal antibodies, at the difference with MLV [59, 60] and after weaning, the DNA-primed piglets could be vaccinated with MLV and they might develop rapid and broad immunity upon transfer to fattening units. Studies of compared efficacy, economic assessments and effectiveness modeling will have to be undertaken in order to bring such PRRSV DNA vaccine component to the field.

Minor comments:

Line 30: agents instead of agent: corrected

Line 55: the instead of he: corrected

Line 116-127: all antibodies described in this section are mAb, so it is unclear why the definition of mAb is given only in line 119 and line 120 states “the mAbs used in this study are…”: now clarified

Fig. 2. The plus and minus signs are not very clear in the table and perhaps could be substituted: modified

Line 281: extra-space after Australia modified

Line 323: extra ) after ml corrected

Line 328: per instead of par corrected

Figure 5. Add a description to the graphs depicted in 5F in the legend. Give meaning of 4 in the legend. corrected

Reviewer 3 Report

This is a well written and presented manuscript, exploring a novel approach to vaccination against PRRS.

The figures are clear and informative, and the conclusions are supported by the data presented.

Minor suggestions for improvement:

1- Throughout the materials and methods section, supplier/manufacturer details are very often missing - please add these.

2- Line 247 - Details of the commercial ELISA could be provided.

3- Line 250 - Was there an acclimatisation period for the animals prior to the start of the study?

4- Lines 537-539 belong in the discussion rather than results section

Author Response

Reviewer 3. Italic = reviewer comment, normal black = our answer, in red = the lane where the modified text is included and the text itself.

This is a well written and presented manuscript, exploring a novel approach to vaccination against PRRS.

The figures are clear and informative, and the conclusions are supported by the data presented.

Minor suggestions for improvement:

1- Throughout the materials and methods section, supplier/manufacturer details are very often missing - please add these. The supplier and manufacturer details are now provided (manufacturer, supplier, city, country).

2- Line 247 - Details of the commercial ELISA could be provided. The information is now provided.

L322: The sera and swab extracts were assayed using the Ingezim PRRS 2.0 kit (Ingenasa, Eurofins-technologies, Bruxelles, Belgium) at a 1:40 and 1:2 dilution respectively. The Ingezim PRRSV 2.0 kit is an PRRSV-1 and 2 nucleocapsid-based indirect ELISA, and present good sensitivity and specificity when compared to other commercial ELISA tests [43].

3- Line 250 - Was there an acclimatisation period for the animals prior to the start of the study? yes, there was a week of acclimation. This information is now provided L247.

4- Lines 537-539 belong in the discussion rather than results section. The sentence has been modified to avoid confusion with a discussion. However, the overall data that we obtained do not permit to really discuss on mucosal immunity aspects.

L540: This finding indicates that the prime-boost strategy, especially with XCR1-targeting of vaccine antigens, promote PRRSV-specific Abs location in mucosa.

Round  2

Reviewer 1 Report

The authors introduced the needed changes. The paper is now fully acceptable.

Reviewer 2 Report

The responses to comments were satisfactory.